# Accelerated Bone Induction of Adult Rat Compact Bone Plate Scratched by Ultrasonic Scaler Using Acidic Electrolyzed Water

**DOI:** 10.3390/ma14123347

**Published:** 2021-06-17

**Authors:** Mamata Shakya, Masaru Murata, Kenji Yokozeki, Toshiyuki Akazawa, Hiroki Nagayasu, Bhoj Raj Adhikari, Chandan Upadhyaya

**Affiliations:** 1Division of Oral Regenerative Medicine, Health Sciences University of Hokkaido, 1757 Kanazawa, Hokkaido 061-0293, Japan; murata@hoku-iryo-u.ac.jp (M.M.); yokozeki@hoku-iryo-u.ac.jp (K.Y.); 2Division of Dental Surgery, Kathmandu University School of Medical Sciences, Dhulikhel 11008, Nepal; bhojrajadhikari@gmail.com (B.R.A.); updch@yahoo.com (C.U.); 3Department of Industrial Technology Research, Hokkaido Research Organization, Sapporo 060-0819, Japan; akazawa-toshiyuki@hro.or.jp; 4Division of Oral and Maxillofacial Surgery, Health Sciences University of Hokkaido, 1757 Kanazawa, Hokkaido 061-0293, Japan; nagayasu@hoku-iryo-u.ac.jp

**Keywords:** acidic electrolyzed water, compact bone, cracks, demineralization, direct bone induction, ultrasonication

## Abstract

Fresh compact bone, the candidate graft material for bone regeneration, is usually grafted for horizontal bone augmentation. However, the dense calcified structure inhibits the release of growth factors and limits cellular and vascular perfusion. We aimed to create mechano-chemically altered dense skull bone by ultrasonic treatment, along with partial demineralization using commercially available acidic electrolyzed water (AEW). The parietal skull bone of an 11-month-old Wistar rat was exposed and continuously treated with a piezoelectric ultrasonic scaler tip for 1 min, using AEW (pH 2.3) or distilled water (DW, pH 5.6) as irrigants. Treated parietal bone was removed, cut into plates (5 × 5 × 1 mm^3^), grafted into the back subcutaneous tissues of syngeneic rats, and explanted at 1, 2, and 3 weeks. AEW bone showed an irregular surface, deep nano-microcracks, and decalcified areas. SEM-EDS revealed small amounts of residual calcium content in the AEW bone (0.03%) compared to the DW bone (0.86%). In the animal assay, the AEW bone induced bone at 2 weeks. Histomorphometric analysis showed that the area of new bone in the AEW bone at 2 and 3 weeks was significantly larger. This new combination technique of AEW-demineralization with ultrasonic treatment will improve the surface area and three-dimensional (3D) architecture of dense bone and accelerate new bone synthesis.

## 1. Introduction

The use of grafts and biomaterials is unremitting in the field of bone surgery. A fresh living bone is a natural composite structure composed of two main components, hydroxyapatite (HAp) and type I collagen. The bone matrix is a storehouse of different types of growth and differentiation factors [1,2]. Owing to these properties, fresh bone has been considered a desirable graft material, and the use of fresh autogenous bone is still regarded as the gold standard, as most biomaterials fail to meet optimal requirements [3,4]. However, it is well known that morbidity at the healthy donor site is the main problem after harvesting fresh bone. 

The ability of bone to repair and remodel itself in response to changing physical demands has been attributed to its inherent property to form and store a number of growth factors. These polypeptides, in very low concentrations, act as local regulators of cell functions. Since the discovery of bone morphogenetic proteins (BMPs), by Urist in 1965 [5], numerous studies have been conducted to investigate the bone-inducing properties [6,7] of various growth factors. While BMPs are known to be the important proteins for bone induction, other growth factors have also been identified in bone, including fibroblast growth factors (FGFs), insulin like growth factors (IGFs), platelet derived growth factors (PDGF), and transforming growth factors-β (TGF-β) [8]. Conventionally, a demineralized bone matrix (DBM) is produced by demineralization outside of the body in hydrochloric acid, nitric acid, or formic acid to efficiently remove minerals [9,10]. Although demineralization in the strong acid solutions such as HCl or HNO_3_ is rapid, the safety of its direct use on living tissue has not been established. Acidic electrolyzed water (AEW), a product of electrolysis of sodium chloride solution, is acidic water at pH 2.2–2.7. AEW contains hypochlorous acid, which is an effective sanitizer at low pH [11]. It is applied in the medical sector and food processing as an oral irrigant and a bactericidal agent [12,13]. More importantly, AEW is readily neutralized after contacting tissues and does not have detrimental effects, which enables AEW to be used directly on living tissues. 

Recently, the scope of ultrasonic modalities has broadened. Ultrasonic treatment at 120 W and 38 kHz for 2 min produces convenient micro-pores and micro-cracks for body fluid permeation [14]. Ultrasonic waves of more than 20 kHz can bring bubble cavitation and form a hot spot. In the hot spot, chemical reactions are activated by the formation of radical groups and locally rising temperatures [15]. The effectiveness of cavitational activity for removing strongly bound particulate matter from solid surfaces is an important property applied for surface modification in bioengineering [16,17]. We hypothesized that an ultrasonic and acid combination technique should modify the cortical bone surface and increase the surface area. 

Owing to the larger effective surface area, a cancellous bone graft is commonly preferred over the compact type. The porous nature of cancellous bone enables easy perfusion of fluid, growth factors, and cells across the graft materials. However, it does not provide adequate biomechanical stability and functional loading [18]. On the contrary, compact a bone graft provides adequate mechanical loading stability and possesses good structural integrity, but limits the free dispersion of body fluids and growth factors because of its highly dense nature. This results in the limited osteogenic and osteoconductive properties of the cortical bone graft [19]. The geometry, biochemical nature, stability, and mechanical strength are important factors determining a suitable graft material. In this study, we developed a newly processed bone, using the physical properties of compact bone combined with the biological properties of cancellous bone. We fabricated a dense plate-type graft derived from the skull bone (11-month-old rat, adult-old stage) with surface modification by ultrasonic power using AEW etching.

This study aimed to analyze the surface modification of skull bone processed by ultrasonically AEW-demineralization and investigate the bone-inductive capability of ultrasonic AEW-treated skull bone in syngeneic young rat subcutaneous tissues.

## 2. Materials and Methods

### 2.1. Materials Used and Division of Groups

#### 2.1.1. Acidic Electrolyzed Water (AEW)

A three-chamber double-in electrolytic system device incorporating an electrolytic apparatus (Redox technology Co., Tokyo, Japan) was used to produce AEW (Figure 1). At 9.1 V and 9.0 A, electrolysis of saturated sodium chloride solution (26.5% NaCl) was continuously and effectively carried out at a flow rate of 4200 cm^3^∙min^−1^. AEW (pH 2.2 to 6.5) with 20–70 mg/kg available chlorine content (ACC) and an oxidation-reduction potential (ORP) of +1100 mV was collected in the anode region of the chamber. AEW at pH 2.3 was used in this study.

#### 2.1.2. Distilled Water (DW)

Bidistilled water of pH 5.6 was used (Otsuka Distilled Water Co., Tokyo, Japan).

#### 2.1.3. Ultrasonic Device

An ultrasonic scaler unit (Piezon Master 700, EMS, Nyon, Switzerland) was operated at a frequency of 24–32 kHz, with an output of 8–12 W, for 1 min using an interdental scaler tip with effective mechanical force (approx. 2 Newton).

#### 2.1.4. Animal Model

Wistar rats (Hokudo Co., Sapporo, Japan) of two different age groups were taken for this study. The donor rats were an 11-month-old female (average weight: 300 g) at adult-old stage, and the recipients were 4-week-old males (average weight: 75 g) at young stage. A total of 45 rats were used in this study. Animal experiments were approved by the Animal Research Center of Health Sciences University of Hokkaido, Japan (authorized no. 109). 

#### 2.1.5. Division of Groups

The graft materials were divided into three groups, depending on the irrigation liquids used. The AEW bone and DW bone groups were treated under AEW and DW irrigation, respectively. The fresh bone group, as a control, was not treated by any irrigants. Among the 45 rats, 5 rats (2 donor and 3 recipient) were used in each group each week. 

### 2.2. Animal Experiment

#### 2.2.1. Preparation of Graft Samples

Donor rats were sacrificed by inhalation of an excessive dose of diethyl ether. A horizontal incision (1 cm) was made at the base of the skull. A full-thickness skin flap, including the periosteum, was reflected to expose the parietal portion. The exposed parietal skull bone surface was treated using a piezo-ultrasonic scaler tip with AEW or DW liquids as irrigants for 1 min. The scaler tip was run perpendicular to the bone surface in a longitudinally uniform direction, starting from the top, to the base of the skull. The scalar tip was operated at 32 kHz, with maximum irrigation and effective mechanical force application. This mechanochemical treatment was performed only on the outer cortex of the skull bone. The skull bone without any treatment was taken as a control and termed the fresh bone group. After the treatment, the parietal bone was excised, washed in saline solution, cut into fragments (5 × 5 × 1 mm^3^), and preserved in sterile and moist condition until grafting (Figure 2). 

#### 2.2.2. Topographical Analysis

Scanning electron microscopy (SEM) was used for topographical analysis of the surface of the treated and non-treated bones before grafting. SEM observation was done to characterize the variation in the surface texture, the presence of microdamage, and the nature of damage present in the three groups. Energy-dispersive X-ray spectroscopy (EDS) is a sensitive qualitative and semi-quantitative technique for evaluating the differences of mineral content in the microscopic region of bone. For each bone treated with different solutions by ultrasonication, the EDS spectra were measured at 10.0 kV and depth regions of 1 μm were analyzed. An SEM-EDS analysis for each bone was made to investigate residual mass % of different elements found in the surface of bone after the treatment. In order to examine the decalcification, calcium to phosphorus (Ca/P) ratio was determined by the EDS analyses.

#### 2.2.3. Subcutaneous Graft in Host Rats

General anesthesia (9 mg/100 g body weight) was used for the recipient rats with intraperitoneal administration of pentobarbital sodium (50 mg/mL). The skin site of the recipients at the posterior abdominal wall was prepared by trimming fur and disinfecting with iodine and 70% alcohol. Vertical incisions (1 cm) were made under sterile conditions on either side of the midline of the body, followed by blunt dissection to form subcutaneous pouches. Previously prepared samples were grafted into this subcutaneous pouch, and the incision was closed with a sterile suture. As the surgical procedures were performed under sterile conditions and minimally invasive, any behavioral changes or any abnormalities were not noted during the 1-h and 24-h follow-up periods. The grafted samples were explanted at 1, 2, and 3 weeks.

#### 2.2.4. Histological Examination

Explanted materials were processed by a standard protocol. Samples were fixed in 10% neutral-buffered formalin solution for 1 week, demineralized in 10% formic acid for 4 weeks, and embedded in paraffin. Prepared paraffin block specimens were sectioned into 5 µm thickness on a microtome (Yamato ROM 308, Tokyo, Japan). Hematoxylin and eosin (HE)-stained sections were observed by an optical microscope (Nikon ECLIPSE 80i, Nikon Corporation, Tokyo, Japan) and photographs were taken.

#### 2.2.5. Histomorphometric Measurements

Histomorphometric analysis was conducted for an area measuring 5 mm in length and 0.5 mm in width, corresponding to the region between the two cut ends and the area cortical from the bone marrow. For each sample, three fields were equidistantly captured at a magnification of 100× under an optical microscope. The area of new bone and the total graft material in each representative field were measured using the Weibel method [20] on the public domain program, ImageJ version 1.50b (National Institute of Health, Bethesda, MD, USA). The total volume of the analyzed area was taken as 100%. The average area of new bone was used as the mean percentage of each group. Data were expressed as the percentage of new bone area compared to that of graft bone.

## 3. Results

### 3.1. SEM Observation and EDS Analysis

#### 3.1.1. Fresh Bone Group

The fresh bone exhibited a smooth surface, likely due to densely arranged organic and inorganic components, along with the presence of physiological cracks (about 5 μm in length, Figure 3a,b). 

#### 3.1.2. DW Bone Group

The DW bone revealed a linear smooth and rough surface structure at lower magnification (Figure 3c). At higher magnification, a densely arranged, irregular, rough, wavy, or scaly bone surface, along with damaged fibers over it, mainly due to the chipping action of scaler tips, along with the microdamage, were observed (Figure 3d,e).

#### 3.1.3. AEW Bone Group

The AEW bone revealed a heterogeneous pattern of damaged smooth and irregular surface areas, along with multiple cracks of more than 50 µm (Figure 3f). The treated bones’ surface resembled a network of damaged fibers with numerous pores (Figure 3g,h). These can be attributed to the running of the scaler tip and chemical demineralization by AEW. 

#### 3.1.4. EDS Analysis

The mean mass % of different elements that remained on the surface of each bone after the treatment is shown in Table 1. The residue calcium % on the surface of bones treated with ultrasonication using AEW and DW was 0.03% and 0.86%, respectively. The phosphorous levels were 0.09% and 1.06% in AEW and DW bones, respectively. The EDS analysis showed that the (Ca/P) ratios were 0.83 for fresh bone, 0.81 for DW bone, and 0.33 for AEW bone, respectively. There were no significant changes in the quantity of mean mass % with respect to carbon, nitrogen, oxygen, sodium, and magnesium. The mean mass % of chlorine was found to be increased in AEW treated bone and decreased in DW treated bone. The changes in chlorine might be attributable to the available chlorine content from the AEW used as an irrigant. 

### 3.2. Histological Findings

#### 3.2.1. Fresh Bone Group

There was no evidence of new bone synthesis until the end of week three (Figure 4). At 1 week, the fresh bone was encapsulated by a few layers of fibrous connective tissue. At 2 weeks, the thickness of the encapsulated fibrous layer had increased without any considerable changes in the graft. At 3 weeks, several shallow excavation pits were observed with multinucleated giant cells on the grafted bone, and enlarged, empty lacunae spaces were seen (Figure 4e,f).

#### 3.2.2. DW Bone Group

At 1 week, the DW bone was encapsulated by a thin layer of connective tissues with only minor morphological changes on the surface (Figure 5a,b). At 2 weeks, the grafted bone was encapsulated by a thick layer of fibrous connective tissues over the layer of osteogenic cells. Multinucleated giant cells were found along the damaged surface. The progression of an excavation chamber on the localized area was evident. Empty lacunae and shrunken nuclei were observed in osteocyte spaces. The presence of large empty lacunae indicated necrosis (dead bone; Figure 5c,d). At 3 weeks, a woven bone formation was locally observed, along with osteoblasts overlaying it. However, the core structure of DW bone remained unaltered. New bone could be easily differentiated from the grafted bone by the presence of a linear cement line between them (Figure 5e,f).

#### 3.2.3. AEW Bone Group

At 1 week, the AEW bone was encapsulated by a thick layer of mesenchymal tissues and fibrous connective tissues. Morphological changes in the grafted bone were observed, with the presence of multiple deep trenches on the dense outer plate. Along with some areas, elongated linear cracks extending into the marrow space were observed (Figure 6a). Multinucleated giant cells were seen in the damaged area (Figure 6b). Empty lacunae spaces were found in the vicinity of the outer cortical plate. At 2 weeks, osteoblasts produced a new bone matrix in the dead bone chamber, and multinucleated giant cells appeared on the irregular outer surface (Figure 6c,d). At 3 weeks, the histological lamellar anatomy of the graft bone was altered. The induced bone could be distinguished from the old dead bone by a cement line between them, and the irregular arrangement of lacunae with osteocytes (Figure 6e,f).

### 3.3. Histomorphometric Analysis

Histomorphometric analysis showed that the amount of new bone in the AEW bone was significantly higher compared to the DW bone at 2 weeks and 3 weeks (*p* < 0.05). New bone occupied approximately 4.2% and 13.4% of the total implant surface in AEW bone at 2 and 3 weeks, respectively, compared to 8.2% in DW bone at 3 weeks. Fresh bone showed 0%, suggesting no bone formation at any point. The amount of new bone increased from week two to week three in the AEW and DW bone (Table 2).

## 4. Discussion

In the present study, we showed that ultrasonic AEW treatment could effectively bring about the surface modification and partial surface demineralization of cortical bone from an adult old-stage rat. Our novel processed bone induced new bone directly over a 2 week period. To the best of our knowledge, this is the first report of rapid new bone synthesis at 2 weeks by a plate-type compact bone. 

Aged bone becomes denser, accompanied by a reduction in a regenerative capacity and a decline in homeostatic mechanisms, creating a long gap between bone loss and regeneration [21,22]. Rats have decreased bone-inductive capacity with aging [23,24]. Ectopic bone formation by DBM in old rats was slower than in younger rats [24]. From the age of 7–8 months, skeletal growth tapers off in male and female Wistar rats [25]. The present study used three types of bone from old adult female rats, and assuming decreased bone induction potential. However, we showed that the safe mechano-chemically-treated compact bone plate derived from 11-month-old female rats could induce new bone formation at 2 weeks post graft in subcutaneous tissues. The formation process might be direct bone formation, without cartilage. Ultrasonic AEW-demineralization can reinforce the potential for bone growth and induce new bone formation.

The geometry, biochemical nature, and mechanical strength of the graft materials are crucial factors for bone regeneration [26,27]. Porous graft materials promote bone formation better than the dense type [28]. Although a plate-type cortical bone provides the necessary support and stability, the cortical bone matrix does not allow for sufficient diffusion of growth factors [3]. Therefore, we focused on the mechano-chemical processing of a dense natured plate-type cortical bone. Ultrasonic scaler treatment with AEW irrigation could bring about surface modification and morphological changes of the hard tissues, such as bone and dentin, in a time-dependent manner [15]. In this study, the SEM results provided evidence that ultrasonic irradiation with AEW enhanced the physical nature of the physiological cracks and the creation of new nano-micro damage, along with attenuating the already present physiological microcracks with destroyed fibers. These microcracks can facilitate migration and proliferation of the mesenchymal cells necessary for bone tissue formation [29]. The changes in the surface morphology correlated strongly with the presence of cuboidal active osteoblast differentiation followed by new bone matrix deposition (Figure 6). 

In addition to nano-microcracks created by ultrasonic devices, the demineralization of bone increases the release and bioavailability of matrix-associated proteins, especially BMPs, thus rendering these grafts osteoinductive [30]. Among the available growth factors FGF, IGF, and TGF play a complementary role, whereby BMP is the main protein known to be osteoinductive [31]. Partially demineralized bone with an optimum amount of calcium remaining shows better bone formation than completely demineralized bone and normal calcified bone [32,33]. Strong acids, such as HCl and HNO_3_, conventionally used for demineralization, are dangerous biohazards to the environment and the operator. The safe use of acids directly on a living body has not been established thus far. On the other hand, AEW is non-corrosive to the skin and mucous membranes, has antimicrobial activity, and is environment friendly [12]. While in vitro studies have shown that immersion of dentin in AEW markedly decreased the microhardness and Ca/P ratio, and increased surface roughness [34], an animal study on the effect of AEW for calcified living bone has not been conducted until now. We assumed that AEW might have similar effects in bone, which closely resembles dentin in its organic and inorganic composition. In the present EDS analysis, AEW bone greatly decreased the remaining mean mass % of Ca and P, compared to the other two groups. The remaining mean mass % of Ca in DW bone was about 28.7 times higher than that in AEW bone, thus leading to partial AEW-demineralization of the superficial layer of the grafted cortical bone. The marked dissolution of calcium and phosphorus and the decreased Ca/P ratio strongly suggested that decalcification occurred. This decreased ratio suggested that AEW has a demineralization ability. The SEM-EDS results supported our assumption that AEW could decalcify the dense bone surface. Thus, the histological and SEM-EDS results support each other. A state of partial demineralization with changes in surface topography and nature could be a factor in the release of matrix associated proteins for new bone induction. In our pilot studies (data not shown), AEW was used as an irrigant in the skull bones of live rats after flap reflection. Normal healing of the periosteum was seen without any detrimental effect on soft tissues. Histological findings have suggested no deleterious effect of AEW on soft tissues. This further proves that AEW is non-corrosive to soft tissues. 

Direct induction of new bone formation was found at 2 weeks in AEW bone, such as intramembranous ossification. BMPs induce direct bone formation and endochondral bone formation, independently [35]. Differentiation of undifferentiated mesenchymal cells is controlled not only by the regulatory factors but also by the cellular environment [35]. We believe, therefore, that direct bone formation might have been caused due to the demineralized superficial area of the cortical bone plate and exposed BMP-binding organic matrix, maintaining the inner structure after ultrasonic scaler treatment with AEW, and releasing matrix-related growth factors from the outer layer into the surrounding cells. The interplay between the growth-related factors, such as Wingless type (Wnt), TGF-β, IGF, parathyroid hormone (PTH), and their receptors with the BMP is a good example of the multitude of crosstalk mechanisms possible. Regulation of BMP by Wnt, PTH, and Osteoprotegerin (OPG) has a significant role in bone formation and resorption cells. Wnt/ β-catenin signaling is known to regulate the differentiation of bone cells from their early precursors [36]. Additionally, BMP bioavailability is regulated by low or high affinity binding to extracellular matrix components [36], which were readily available due to the partial demineralization by AEW in the present study. The detailed mechanism of bone formation is beyond the scope of this study. It was hypothesized that the combination of physiochemical remineralization and osteocyte-mediated processes might play some role in this kind of repair [37]. In addition to the surface microdamage created by the ultrasonic power, the wave generated may affect the entire thickness of the graft. This may trigger the lacuna-canalicular network of osteocytes, thus, modulating regulatory molecules involved in bone formation [38]. The direct formation of new bone over the surface of old bone, without undergoing a remodeling process has been termed mini-modeling [39]. The present results in both the AEW and DW bone groups support the mini-modeling concept. Further experiments are needed to understand the mechanism of bone formation.

## 5. Conclusions

The ultrasonically demineralized AEW bone plate had a better performance for the induction of new bone formation than fresh bone and DW bone. We concluded that this novel technique, using the combination of an ultrasonic scaler tip and AEW (pH 2.3), could create nano-micro cracks on the highly-dense cortical bone and accelerate the induction of new bone formation.

## Figures and Tables

**Figure 1 materials-14-03347-f001:**
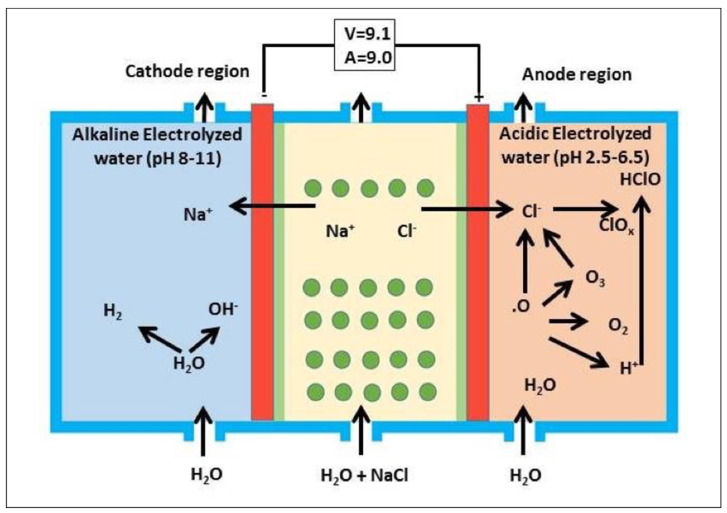
A three-chamber double-in electrolytic system device incorporating an electrolytic apparatus (Redox Technology Ltd. Co., Tokyo, Japan).

**Figure 2 materials-14-03347-f002:**
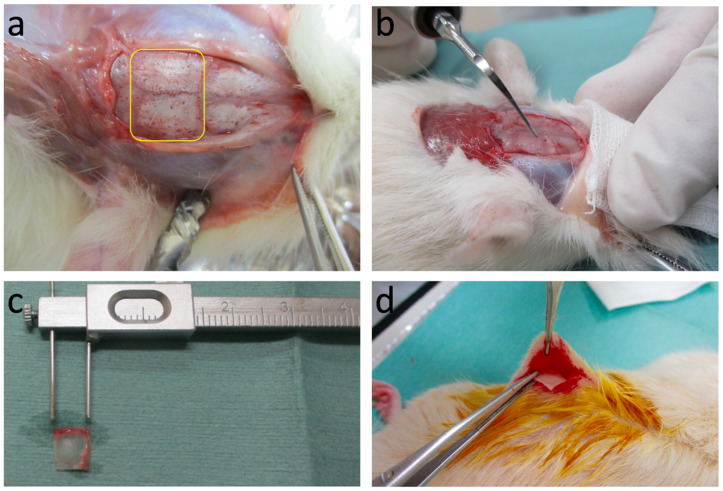
Preparation of graft material. (**a**) Exposure of parietal bone (yellow box) after reflection of full thickness skin flap. (**b**) Exposed surface treated by piezo-ultrasonic scaler tip using AEW or DW irrigant. (**c**) Treated bone prepared on plate size (5 × 5 × 1 mm^3^). (**d**) Placement of prepared graft into subcutaneous pouch of recipient rat.

**Figure 3 materials-14-03347-f003:**
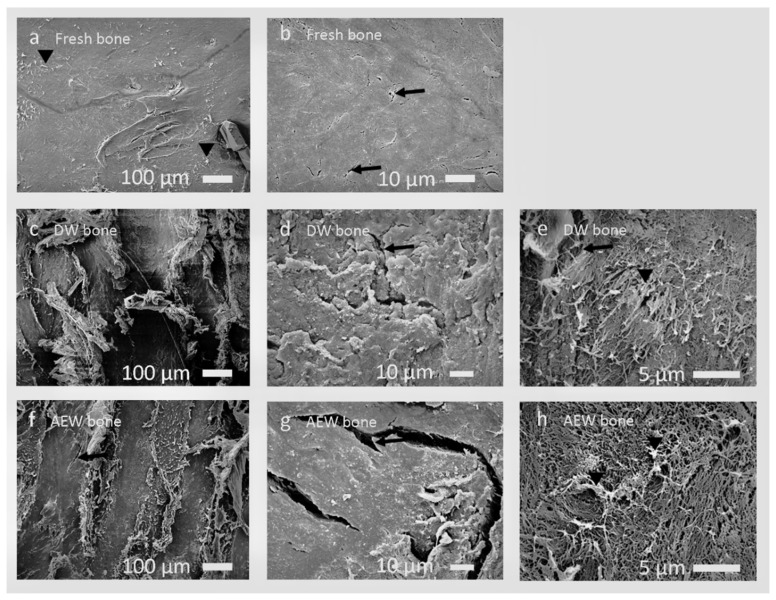
SEM photographs of fresh bone group, without any treatment (**a**,**b**), DW bone group (**c**–**e**), and AEW bone group (**f**–**h**). Fresh bone group: (**a**) smooth clear surface with few destroyed fibers (▾), (**b**) physiological cracks around 5 µm in length (arrow). DW bone group: (**c**) linear smooth and rough surface structure, (**d**) rough scaly bone surface with micro damage or linear cracks (arrow), (**e**) densely arranged microstructure with destroyed fibers (▾) along with linear cracks (arrow). AEW bone group: (**f**) heterogeneous smooth and irregular surface structure with multiple cracks (arrow), (**g**) distinct linear cracks (arrow), (**h**) network of damaged fibers (▾).

**Figure 4 materials-14-03347-f004:**
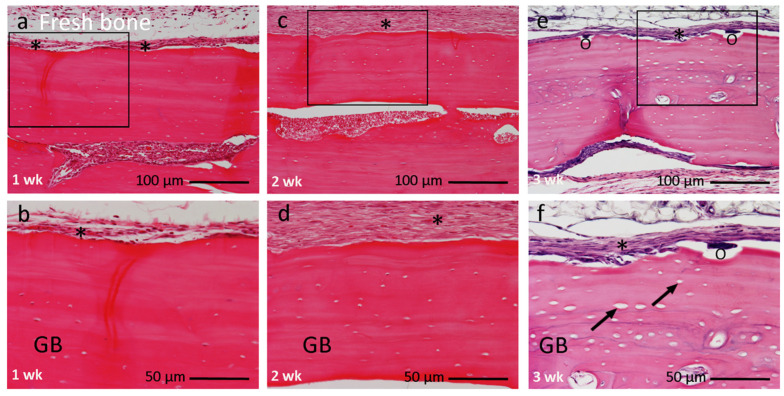
Histological image of the fresh bone group at 1, 2, and 3 weeks after implantation with higher magnification. (**a**,**b**) At 1 week graft bone encapsulated by few layers of connective tissue (*). (**c**,**d**) At 2 weeks, graft bone encapsulated by a thick layer of fibrous connective tissue (*). (**e**,**f**) At 3 weeks, shallow excavation pit with multinucleated giant cells (O). No new bone formation. Boxes within a, c, e indicate focused area in (**b**,**d**,**f**), respectively. (**a**,**b**) 1 week, (**c**,**d**) 2 weeks, (**e**,**f**) 3 weeks. Magnification: (**a**,**c**,**e**) 10×, (**b**,**d**,**f**) 20×. (↗) = empty lacunae. Bar: (**a**,**c**,**e**) 100 µm; (**b**,**d**,**f**) 50 µm.

**Figure 5 materials-14-03347-f005:**
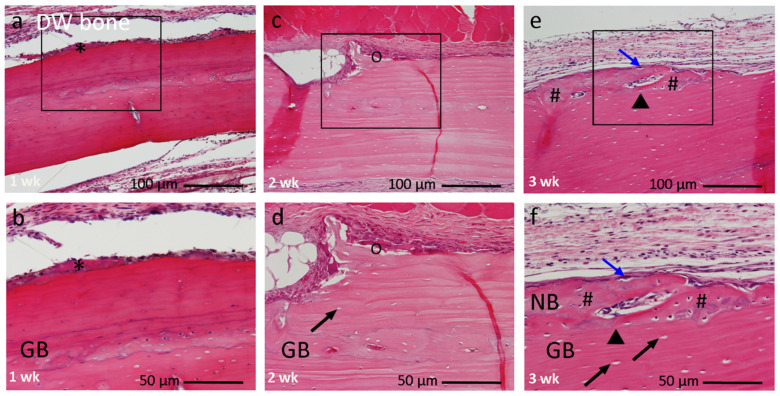
Histological image of DW bone group at 1, 2, 3 weeks after implantation, at higher magnification. (**a**,**b**) At 1 week grafted bone encapsulated by a thin layer of connective tissue (*). (**c**,**d**) At 2 weeks progression of excavation chamber with multinucleated giant cells (O). (**e**,**f**) At 3 weeks localized woven bone induction (#) along outer cortical plate clearly separated by a cement line (▾). Box within a, c, e indicates the focused areas in (**b**,**d**,**f**), respectively. (**a**,**b**) 1 week, (**c**,**d**) 2 weeks, (**e**,**f**) 3 weeks. Magnification: (**a**,**c**,**e**) 10×, (**b**,**d**,**f**) 20×. (↗) = empty lacunae, (↘) = early active cuboidal osteoblast cell differentiation, GB: graft bone, NB: new bone. Bar: (**a**,**c**,**e**) 100 µm; (**b**,**d**,**f**) 50 µm.

**Figure 6 materials-14-03347-f006:**
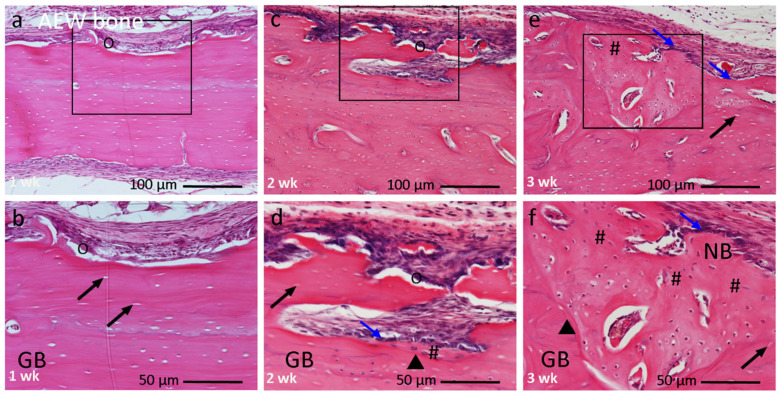
Histological image of AEW bone group at 1, 2, and 3 weeks after implantation, at higher magnification. (**a**,**b**) At 1 week graft bone encapsulated by a thick layer of mesenchymal connective tissue. Presence of trenches on outer cortical plate with multinucleated giant cells (O). (**c**,**d**) At 2 weeks resorption of damaged old bone. Cuboidal active osteoblast differentiation (↘) followed by new bone induction (#). (**e**,**f**) At 3 weeks abundant new bone induction (#) along original dead bone with irregular arrangement of new osteocytes, clearly separated by a cement line (▾). Box within a, c, e indicates focused areas in (**b**,**d**,**f**), respectively. (**a**,**b**) 1 week, (**c**,**d**) 2 weeks, (**e**,**f**) 3 weeks. Magnification: (**a**,**c**,**e**) 10×, (**b**,**d**,**f**) 20×. (↗) = empty lacunae, GB: graft bone, NB: new bone. Bar: (**a**,**c**,**e**) 100 µm; (**b**,**d**,**f**) 50 µm.

**Table 1 materials-14-03347-t001:** Elemental analysis values of SEM-EDS spectra for skull parietal bone irradiated ultrasonically with different solutions.

Element	Mean Peak Intensity % ± SD		
	NS	DW	AEW
Carbon	42.44	45.85	45.21 ± 1.95
Nitrogen	37.69	37.72	38.55 ± 0.53
Oxygen	17.61	14.25	15.66 ± 2.05
Sodium	0.14	0.20	0.12 ± 0.10
Magnesium	0.03	0.02	0.01 ± 0.00
Phosphorus	1.08	1.06	0.09 ± 0.01
Chlorine	0.10	0.00	0.32 ± 0.05
Calcium	0.90	0.86	0.03 ± 0.00
Total	99.99	99.96	99.99

SEM-EDS spectra of skull parietal bone treated with different solutions were measured at 10.0 kV. Remaining mean mass % of different elements were found on the surface of bone after ultrasonic treatment with three different solutions. NS, normal saline; DW, distilled water; AEW, acidic electrolyzed water (n = 2).

**Table 2 materials-14-03347-t002:** Histomorphometric analysis of induced bone at 1, 2, and 3 weeks.

Bone Type	Explant Period		
	1 wk	2 wk	3 wk
Fresh bone (%)	0	0	0
DW bone (%)	0	0	8.2 ± 1.0 **
AEW bone (%)	0	4.2 ± 0.4 **	13.4 ± 1.3 **

Fresh bone showed no bone induction at any time period. New bone was significantly higher in AEW bone compared to DW bone at 2 and 3 weeks (n = 5, ** *p* < 0.05, Mann–Whitney U test). Total volume of the analyzed area: 100%.

## Data Availability

Not applicable.

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
