# Peer review of "Accelerated Bone Induction of Adult Rat Compact Bone Plate Scratched by Ultrasonic Scaler Using Acidic Electrolyzed Water"

_materials, 2021, doi:10.3390/ma14123347_

Round 1

Reviewer 1 Report

It is a well-written paper concerning the influence of ultrasound in the presence of acidic electrolyzed water onto the properties of the rat bones. The obtained data are analyzed with SEM-EDX, hysltological and histomorphometric examinations. To the best of the authors' knowledge, this is the first report of rapid bone induction.
Introduction is written in a comprehensive good manner giving the readers the basics of bone grafting, bone content, use of acidic electrolyzed water (AEW) with pH = 2-3 and ultrasound for the bone surface treating, etc. Authors logically assumed that the combined application of ultrasound and AEW should give some new interesting effects, like improving surface area as well as 3D structure of the material. It is quite important because the porous nature of cancellous bone enables easy perfusion of fluid, growth factors, cells across the graft materials, etc., but the mechanical properties (stability) should also do not suffer in these treatments. The main goal of the paper was to investigate the rat bones under the AEW-assisted ultrasound influence by the methods which the researchers have at their disposal.
Section Materials and Methods is written very clear and describes the applied methods and materials in details. Section Results shows comparative measurements by the methods applied for the fresh bones, AEW and distilled water treated bone groups. In section Discussion an attempt to understand the obtained results is presented. From the data the quite logical conclusions are made that the applied by the authors treatment procedure leads to the better bone induction.
There are some minor comments which could help to improve the paper but they are not crucial for the understanding the results of the manuscript and the whole story line. Therefore, I do no insist on even some minor changes of the manuscript and, in my opinion, the paper can be published in Materials as it is.

Author Response

Dear Reviewer 1

Thank you very much for the time and effort that you have devoted towards the current manuscript. I have made few changes in the manuscript as suggested by other reviewers. I look forward to your cooperation and acceptance of those changes made. The manuscript has been submitted as marked-up version for easy viewing of the changes and another clean version is also attached with it.

Reviewer 2 Report

Shakya and colleagues present an interesting article comparing the effects of the pretreatment of parietal bone of Wistar rats with distilled water (DW) or acidic electrolyzed water (AEW) in the formation of new bone through 3 weeks after syngeneic graft.

Although the article is well written and explained, some improvements and explanations have to be done before accepting the article.

Mayor points:

  1. Authors should explain why they have chosen different genders for the donors (female) and for the grafted (male). Have authors done the same experiences in the other way around. Moreover, authors should explain why donors are “aged rats” and grafted animals “young rats”. Again, the same question did authors try to do the experiences in the other sense? It seems more logical to graft aged animals with bone from young donors. Authors should explain this point.
  2. How many rats were used in each group? Did authors observe any differences in the way rats where eating after chirurgical intervention? Did rats receive any treatment against pain after the intervention? Authors should have realized at least some serum dosages, for example for example to evaluate inflammation in the different groups.
  3. Authors should better explain in material and methods section why they are going to realize EDS.
  4. One important point that is missing is the evaluation of calcification in each type of bone. Authors have calculated the % of some chemical elements, among them Ca and P. Authors should realize other experiences to better evaluate and quantify mineralization and bone formation in each type of bone. Authors should quantify calcium content with o-cresophtalein complex assay which is easy (K. Lorentz 1982 Clin Chim Acta 126 : 327-334). Authors can also evaluate TNAP activity in bone sections or realize Von Kossa staining.
  5. Authors should better explain results from SEM-EDS; How many samples have been evaluated? If there are several samples, authors should present the results in another way (value +/- SD for example). Authors should comment the differences in carbon, oxygen and chlorine content in each group.
  6. Authors should discuss other possible factors that could stimulate bone formation and that can be trapped in bone matrix as Wnts, PTHrP or osteoprotegerin for example.

Minor points:

  1. Please define SEM-EDS in the abstract.
  2. Please replace the term “bone induction” throughout all the article with something more accurate like “new bone synthesis” or “induction of new bone formation” for example.
  3. Please explain in the introduction with some examples why bone is the desirable graft material.
  4. Please explain how distilled water is obtained (mono- or bidistillation).
  5. Legends text seems to be mixed with general text through all the article, please correct.
  6. Latin expression, ex: in vitro, has to be written in italics.

Author Response

Dear Reviewer 2

Thank you very much for the time and effort that you have devoted towards the current manuscript. I have taken the suggestions and questions positively. I have put an attempt to address your questions as much as possible and have adjusted the manuscript to the highest possible extent. However, I deeply apologize that I am unable to completely address some of the points raised by the reviewer, mainly because of the change in my working location, and unavailability of the experimental animals in my present research unit. Attached, please find my response for the queries raised. The manuscript has been submitted as marked-up version for easy viewing of the changes.

  • Mamata Shakya (Corresponding Author)

Reviewer 3 Report

This study is very well design and manuscript is arranged in a very well manner. I highly appreciate the scientific progress made in this research. The only suggestion from myside is to arrange the SEM photographs in a better manner so that readers can pick up the difference in one glance. The rest is good.

Here are some minor changes for the manuscript.

  1. The abstract shall be shorten as in present form it is hard to grab the concept of paper and will demotivate the reader.
  2. . The conclusion is no included in the manuscript. In my view conclusion is an integral part of manuscript. Please include it.
  3.  The histopathological photographs are good but to include a table abiut the difference in different sample will enahnce the manuscript overall interest.
  4. I will also appreciate if the SEM ohotographs are organized in a amnner ao that reader can easily grab the differences. 
  5. If possible please further elaborate the anumal protocol,  so that it is easy to reproduce.

Author Response

Dear Reviewer 3

Thank you very much for the time and effort that you have devoted towards the current manuscript. I have taken the suggestions and questions positively. I have put an attempt to address your questions as much as possible and have adjusted the manuscript to the highest possible extent. However, I deeply apologize that I am unable to completely address some of the points raised by the reviewer, and I have attached the explanation for the same. I look for your cooperation. Attached, please find my response for the queries raised. The manuscript has been submitted as marked-up version for easy viewing of the changes and another clean version is also attached with it.

  • Mamata Shakya (Corresponding Author)

  1. The abstract shall be shorten as in present form it is hard to grab the concept of paper and will demotivate the reader.

Answer: Thank you for the suggestion. We have summarized the study in the abstract, however, due to word count limitation (limited to 200 words in abstract), we were not able to include some of the points.

  1. . The conclusion is no included in the manuscript. In my view conclusion is an integral part of manuscript. Please include it.

Answer: We have added the conclusion section.

  1.  The histopathological photographs are good but to include a table about the difference in different sample will enhance the manuscript overall interest.

Answer: We have explained the differences in the result section and compared the findings between different time periods in respective areas, although it is not presented in tabulated form.

  1. I will also appreciate if the SEM photographs are organized in a manner so that reader can easily grab the differences.

Answer: Thank you very much for the suggestions. We have addressed this point and readjusted the picture.

  1. If possible please further elaborate the animal protocol,  so that it is easy to reproduce.

Answer: Thank you very much for the suggestion. We have used standard protocols in surgical approaches to the animals and explained in the manuscript accordingly.

Round 2

Reviewer 2 Report

Authors have not addressed some of the major points that have been suggested, notably the points from 3 to 6:

3. Authors should better explain in material and methods section why they are going to realize EDS.

Authors claim they don't have enough place but at least one phrase can be added in material and method section.

4. One important point that is missing is the evaluation of calcification in each type of bone. Authors have calculated the % of some chemical elements, among them Ca and P. Authors should realize other experiences to better evaluate and quantify mineralization and bone formation in each type of bone. Authors should quantify calcium content with o-cresophtalein complex assay which is easy (K. Lorentz 1982 Clin Chim Acta 126 : 327-334). Authors can also evaluate TNAP activity in bone sections or realize Von Kossa staining.

Authors have not answered this point. authors should realize at least one other experiment to demonstrate the mineralization.

5. Authors should better explain results from SEM-EDS; How many samples have been evaluated? If there are several samples, authors should present the results in another way (value +/- SD for example). Authors should comment the differences in carbon, oxygen and chlorine content in each group.

Authors have only realized experiences twice. Authors should correct results and show the mean+/- SD.

6. Authors should discuss other possible factors that could stimulate bone formation and that can be trapped in bone matrix as Wnts, PTHrP or osteoprotegerin for example.

Authors should improve discussion section adding bibliography about Wnts, PTHrP or osteoprotegerin, which are trapped in bone matrix, and are important to stimulate bone formation.

Author Response

Dear Reviewer 2

(Round 2)

Once again, thank you very much for the time and effort to recheck the manuscript and answers presented in 1st round of review. I have attempted to address the additional suggestions presented by you. Attached, please find my response for the queries raised. The manuscript has been submitted as marked-up version with green fonts for the 2nd round of review adjustments.

  • Mamata Shakya (Corresponding Author)

  1. Authors should better explain in material and methods section why they are going to realize EDS. Authors claim they don't have enough place but at least one phrase can be added in material and method section.

Answer: We have adjusted the manuscript to address the above-mentioned point at “Materials and Methods” sections. Further explanations to justify the queries have been given and mentioned in “results” and “discussion” sections as well.

  1. One important point that is missing is the evaluation of calcification in each type of bone. Authors have calculated the % of some chemical elements, among them Ca and P. Authors should realize other experiences to better evaluate and quantify mineralization and bone formation in each type of bone. Authors should quantify calcium content with o-cresophtalein complex assay which is easy (K. Lorentz 1982 Clin Chim Acta 126 : 327-334). Authors can also evaluate TNAP activity in bone sections or realize Von Kossa staining.

Authors have not answered this point. authors should realize at least one other experiment to demonstrate the mineralization.

Answer: Thank you for your extremely valuable suggestion. We have noted your comments and suggestions and realized its value in this manuscript. We have quantified the new bone formed in each group by using histomorphometric analysis. We are extremely sorry that we are unable to perform the additional experiment to do quantitative analysis for the amount of calcium in processed bone after implantation. We will incorporate this in our future studies.

  1. Authors should better explain results from SEM-EDS; How many samples have been evaluated? If there are several samples, authors should present the results in another way (value +/- SD for example). Authors should comment the differences in carbon, oxygen and chlorine content in each group.

Authors have only realized experiences twice. Authors should correct results and show the mean+/- SD.

Answer: We have explained that there were no significant changes in the mean mass % of minor elements. This has been explained in the result and discussion sections, necessary adjustments have been made.

The data obtained from SEM-EDS analysis as presented before-hand are the mean value. Necessary adjustments have been made in respective sections. The value of AEW bone are now presented as mean ± SD. The values for fresh bone and DW bone were very close to each other with SD value in the range of approx 0.01-0.001, and therefore we could not present as mean ±  SD value.

  1. Authors should discuss other possible factors that could stimulate bone formation and that can be trapped in bone matrix as Wnts, PTHrP or osteoprotegerin for example.

Authors should improve discussion section adding bibliography about Wnts, PTHrP or osteoprotegerin, which are trapped in bone matrix, and are important to stimulate bone formation.

Answer: The above mentioned points are included in the manuscript in discussion section; and explanations of BMP and other growth factors with their possible interaction in bone formation are given as necessary. However, study and analysis of detail mechanism of these factors and the associated pathway are beyond the scope of the present study. Therefore additional experiment cannot be done at present.

Round 3

Reviewer 2 Report

Authors should check Figure 3 because legend and text are still mixed. Tables take too much place, they have to be in line with the text. Discussion is still poor, authors persist to talk only about BMPs, there are a huge number of growth factors trapped in matrix which induce bone formation and this should be addressed in at least 3-4 lines with 1 or 2 references in the discussion section. 

Author Response

Dear Reviewer 2

(Round 3)

Once again, thank you very much for the time and effort to recheck the manuscript and answers presented in 1st and 2ndround of reviews. Your suggestions and recommendations have added the value of this manuscript to the readers. I have attempted to address the additional suggestions presented by you. Attached, please find my response for the queries raised. The manuscript has been submitted as marked-up version with blue fonts for the 3rd round of review adjustments.

  • Mamata Shakya (Corresponding Author)

Comments: Authors should check Figure 3 because legend and text are still mixed. Tables take too much place, they have to be in line with the text. Discussion is still poor, authors persist to talk only about BMPs, there are a huge number of growth factors trapped in matrix which induce bone formation and this should be addressed in at least 3-4 lines with 1 or 2 references in the discussion section. 

Answer: Thank you for your suggestion.

Text mix-up in figure 3 have been corrected. We have checked and addressed similar mistakes in other figures and tables as well.

We have attempted to address your suggestion for BMP and made necessary addition along with citations.
